# Style-Aligned Image Composition for Robust Detection of Abnormal Cells in Cytopathology

**Qiuyi Qi**[*1]                                      QIQIUYI@ZJU.EDU.CN
**Xin Li**[*2,3,5]                                     MERLEY@MAIL.USTC.EDU.CN
**Ming Kong**[1]                                       ZJUKONGMING@ZJU.EDU.CN
**Zikang Xu**[2,3]                                      ZIKANGXU@MAIL.USTC.EDU.CN
**Bingdi Chen**[4,5]                                    INANOCHEN@TONGJI.EDU.CN
**Qiang Zhu**[†1]                                       ZHUQ@ZJU.EDU.CN
**S Kevin Zhou**[†2,3]                                  SKEVINZHOU@USTC.EDU.CN

[1] *Zhejiang University, Hangzhou 310058, China*

[2] *School of Biomedical Engineering, Division of Life Sciences and Medicine, University of Science and Technology of China (USTC), Hefei Anhui, 230026, China*

[3] *Center for Medical Imaging, Robotics, Analytic Computing & Learning (MIRACLE), Suzhou Institute for Advance Research, USTC, Suzhou Jiangsu, 215123, China*

[4] *The Institute for Biomedical Engineering & Nano Science, Tongji University School of Medicine, Shanghai, 200331, China*

[5] *Zhihui Medical Technology (Shanghai) Co., Ltd., Shanghai, 200333, China*

**Editors:** Accepted for publication at MIDL 2025

## Abstract

Challenges such as the lack of high-quality annotations, long-tailed data distributions, and inconsistent staining styles pose significant obstacles to training neural networks to detect abnormal cells in cytopathology robustly. This paper proposes a style-aligned image composition (SAIC) method that composes high-fidelity and style-preserved pathological images to enhance the effectiveness and robustness of detection models. Without additional training, SAIC first selects an appropriate candidate from the abnormal cell bank based on attribute guidance. Then, it employs a high-frequency feature reconstruction to achieve a style-aligned and high-fidelity composition of abnormal cells and pathological backgrounds. Finally, it introduces a large vision-language model to filter high-quality synthesis images. Experimental results demonstrate that incorporating SAIC-synthesized images effectively enhances the performance and robustness of abnormal cell detection for tail categories and styles, thereby improving overall detection performance. The comprehensive quality evaluation further confirms the generalizability and practicality of SAIC in clinical application scenarios. Our code will be released at https://github.com/Joey-Qi/SAIC.

**Keywords:** Cytopathological Diagnosis, Abnormal Cell Detection, Data Augmentation, Image Composition

## 1. Introduction

Due to its non-invasive, efficient, convenient, and cost-effective advantages, abnormal cell identification from cytopathological images has been widely applied in clinical diagnostics.

---

* Equally contributed to this work.
† Corresponding author.

However, traditional cytopathological screening, such as the ThinPrep Cytology Test (TCT) relies on human experts to identify abnormal cells from gigapixel whole slide images (WSI), which is time-consuming, tedious, and heavily dependent on the subjective expertise of pathologists, posing challenges on scarce and imbalanced medical resources (Pan et al., 2024). Consequently, introducing deep learning methods to address abnormal cell detection in pathological images to improve efficiency and reduce the workload of physicians has become a prominent research focus (Yin et al., 2024).

Collecting training sets for abnormal cell detection models faces numerous issues. First, curating a large-scale and accurate dataset needs highly specialized expertise and substantial labor costs. Second, the imbalanced abnormal cell distribution exhibits a typical long-tailed characteristic, leading to performance and robustness concerns in tail categories. Lastly, pathological images acquired from different institutions or periods often exhibit significant differences in staining styles and image qualities, requiring robustness to handle the variation and diversity in real-world scenarios.

One mainstream approach to addressing these issues is to introduce data augmentation. Traditional methods like affine transformations or noise addition increase the diversity of geometric and local perturbations. However, they fail to effectively address the lack of diversity in long-tailed categories and staining styles. With the advancement of GANs (Goodfellow et al., 2020) and diffusion models (Ho et al., 2020), generation-based data augmentation has gained significant attention in pathological image analysis. For example, (Hou et al., 2019) proposed a GAN-based hybrid synthesis pipeline for generating pathological images using predefined rules and textures. (Shen et al., 2024) explored applying parameter-efficient fine-tuning (PEFT) techniques to customize diffusion models for synthesizing cervical cytopathological images. Despite their effectiveness, these methods require fine-tuning, and thus remain constrained by data scale, distribution, and style biases. Another mainstream strategy is to produce augmented data by composing existing foregrounds and backgrounds. For example, Paint-by-Example (Yang et al., 2023) and ObjectStitch (Song et al., 2023) use CLIP (Radford et al., 2021) image encoder to convert the foreground image as an embedding for guidance, thus painting a semantic consistency object on the background image. However, these methods are not specially designed for pathological diagnosis and exhibit deficiencies in style consistency and fidelity.

This paper proposes a novel training-free **S**tyle **A**ligned **I**mage **C**omposition (SAIC) framework, which seamlessly "injects" abnormal cells into specified locations of pathological images while ensuring high fidelity and consistency in categories, types, areas, and staining styles between the foreground and background. Specifically, SAIC consists of three steps: (1) **Attribute-based selection**: Using prior knowledge of category, type, and area to select candidate abnormal cells from an existing cell bank; (2) **Style-aligned composition**: Performing online staining style alignment by reconstructing high-frequency details between the abnormal cell candidates and the style reference image to ensure style consistency in the synthesized area; (3) **LVLM-based filtration**: Leveraging a large visual-language model (LVLM) to filter high-quality samples from synthesized pathological images. Experimental results demonstrate that SAIC achieves high-fidelity, style-preserving data augmentation, effectively enhances the detection of tail categories and rare styles of abnormal cells, and improves overall cell detection performance.

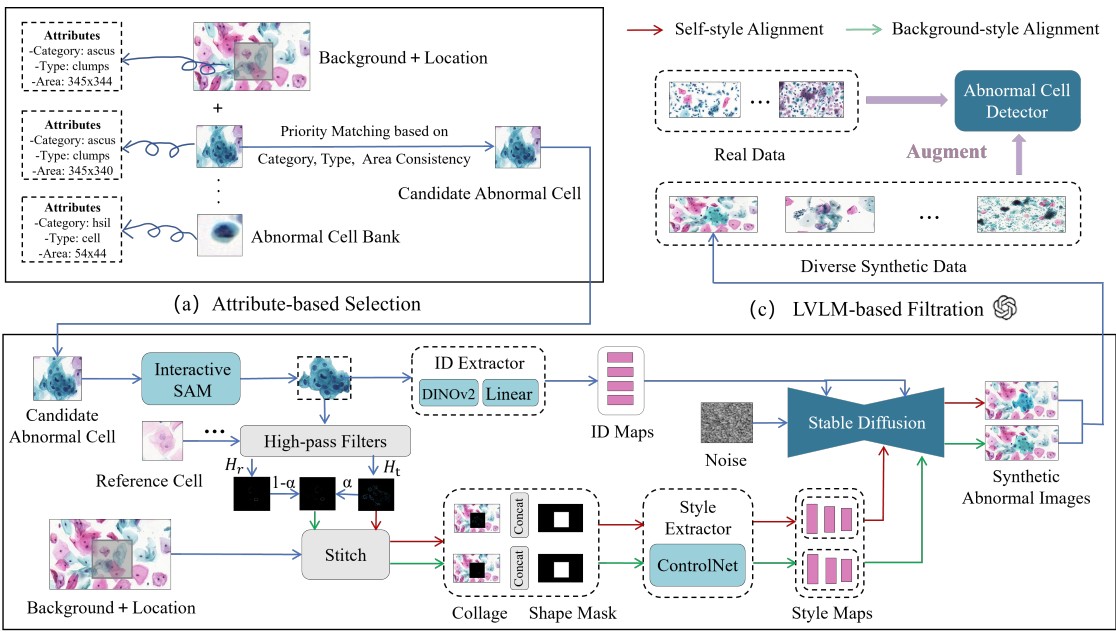

Figure 1: Overall pipeline of SAIC.

We summarize the contributions of this paper as follows: (1) We propose an image composition-based data augmentation architecture for cytopathological abnormal cell detection. (2) Compared to existing generation-based data augmentation methods, the proposed Style Aligned Image Composition (SAIC) framework offers advantages such as being training-free, style-aligned, and high-fidelity. (3) Extensive experimental results demonstrate that SAIC-synthesized images effectively improve abnormal cell detection performance, particularly for tail categories and rare staining styles, while with high-fidelity.

## 2. Method

Figure 1 illustrates the overall pipeline of SAIC. First, **Attribute-based Selection** selects a candidate abnormal cell from an abnormal cell bank based on category, type, and area consistency. Then, **Style-aligned Composition** extracts ID maps and style maps, parallelly producing two synthetic images with self- and background-style alignment. Finally, **LVLM-based Filtration** picks the image with higher fidelity as the final output.

### 2.1. Attribute-based Selection

We first create an abnormal cell bank $\mathcal{C}_{Bank}$ that contains various abnormal cells with region and attribute annotations, which can be easily acquired from the labeled training set. Given a background of cytopathological image $B$ and the target region $L$ with the original cell $c_{orig}$, we select a candidate abnormal cell $c_{cand}$ for composition from the cell bank $\mathcal{C}_{Bank}$ according to the category $\hat{m}$, type $\hat{t}$ (cell/clumps) and area size $\hat{a}$ of $c_{orig}$:

$$c_{cand} = \arg\min_{c \in \mathcal{C}_{Bank}} |a_c - \hat{a}|$$
$$s.t. \ m_c = \hat{m} \text{ and } t_c = \hat{t} \tag{1}$$

where $m_c$, $t_c$, and $a_c$ represent the abnormal cell $c$'s category, type, and area, respectively.

### 2.2. Style-aligned Composition

As the candidate abnormal cell is selected, we must compose it with the background while preserving identity and style consistency. To this end, we design a three-step process:

**Identity map extraction.** For identity preservation, we extract discriminative identity features from the abnormal cell as an ID map. First, we segment the cell region with an interactive SAM (Kirillov et al., 2023) and center-align it. Then extract the visual feature with a DINOv2 encoder (Oquab et al., 2023), followed up with a visual-to-text linear projection to get the ID maps $\mathcal{F}_{ID}$.

**Style map extraction.** For style preservation, we extract the style map with high-frequency information to incorporate style-aware detail guidance. Specifically, first, we select a style reference cell $c_{ref}$ from the abnormal cell bank $\mathcal{C}_{Bank}$ to provide high-frequency information akin to the background's staining style. The selection process is denoted as:

$$S(c_{orig}, c) = \frac{\text{DINOv2}(c_{orig}) \cdot \text{DINOv2}(c)}{\|\text{DINOv2}(c_{orig})\| \|\text{DINOv2}(c)\|},$$
$$C_{ref} = \arg\max_{c \in C_{bank}} S(c_{orig}, c) \tag{2}$$

Subsequently, we extract high-frequency maps $H_t$ and $H_r$ for the candidate abnormal cell and the reference cell, respectively by High-pass Filters (Kanopoulos et al., 1988). Considering the style preservation of both the candidate cell and background, we parallelly extract two kinds of high-frequency maps for self- and background-style alignment:

$$H_n = \begin{cases} H_t, & \text{self-style alignment.} \\ \alpha \cdot H_t + (1 - \alpha) \cdot H_r, & \text{background-style alignment} \end{cases} \tag{3}$$

In practice, the reconstruction coefficient $\alpha$ is empirically set to 0.1 to keep the most important textual information of the candidate cell.

Finally, we stitch both kinds of high-frequency maps into the background and concatenate them with the shape mask of the composition location as inputs of a ControlNet (Zhang et al., 2023) to generate hierarchical style maps $\mathcal{F}_{style}$.

**Conditioned Composition.** Injecting the ID map $\mathcal{F}_{ID}$ and style map $\mathcal{F}_{style}$ into a Stable Diffusion model (Rombach et al., 2022) to guide the composition. Specifically, $\mathcal{F}_{ID}$ are integrated via cross-attention at each UNet layer for identity preservation, while $\mathcal{F}_{style}$ are concatenated with decoder features at each resolution for style preservation. The latent representation of the synthesized image is generated as follows:

$$\mathbf{z}_t = \alpha_t \hat{\mathbf{x}}_\theta(\epsilon, \mathcal{F}_{ID}, \mathcal{F}_{style}) + \sigma_t \epsilon, \tag{4}$$

where $\alpha_t$ and $\sigma_t$ are denoising hyperparameters, which stay aligned with the setting of Stable Diffusion.

All the aforementioned models, including the interactive SAM segmenter, DINOv2-based encoder and its Linear layer, ControlNet, and Stable Diffusion, directly apply the parameters pre-trained on general datasets and do not need specific domain fine-tuning.

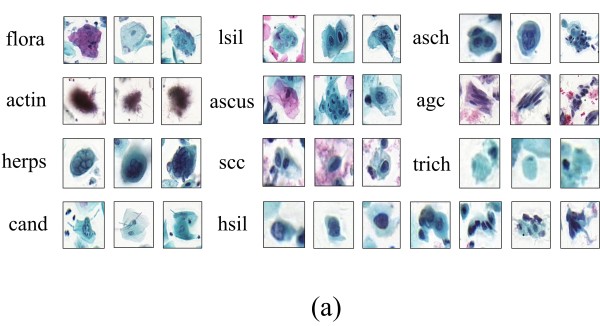
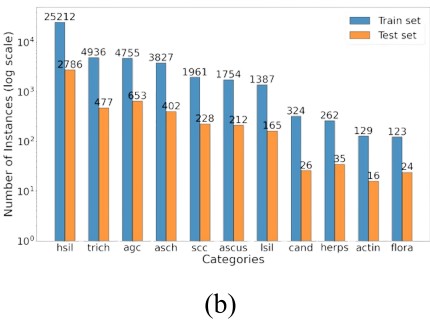

(a)          (b)

Figure 2: Overview of Comparison detector Database. Figures (a) and (b) respectively demonstrate examples of each category and their distribution in the dataset.

## 2.3. LVLM-based Filtration

As we produce two synthetic images with self- and background-style aligned in parallel (See Appendix A for examples), we need to keep the more harmonized one as the final output to raise the augmented data quality. We introduce an LVLM (GPT-4) to make the choice. To mitigate the potential existing positional bias of VLMs, we systematically shuffled the order of the two choices during the experiments. The detailed prompt setting of filtration and the LVLM's filtration ratio of the two styled images are shown in Appendix B.

## 3. Experiment

### 3.1. Datasets and Settings

**Dataset.** We conduct experiments on the Comparison Detector Database (Liang et al., 2018), the largest public dataset for cervical cancer cell detection. This dataset comprises 7,410 cervical microscopic images with 50,447 abnormal cells across 11 categories, as illustrated in Figure 2. We define the categories with fewer than 500 cells as tail categories, while the remainder are non-tail categories. The abnormal cell bank $\mathcal{C}_{Bank}$ is composed of 824 randomly selected abnormal cells, where each category includes 68-90 samples.

**Implementation details.** We evaluate data augmentation methods using two object detectors: YOLOv8 (Varghese and Sambath, 2024) and Faster R-CNN (Ren, 2015). Models are trained using the SGD optimizer (Ruder, 2016) with an initial learning rate of 0.01, a momentum of 0.937, a weight decay of 0.0005, and a batch size of 8, for 150 epochs.

**Evaluation metrics.** We use mAP to evaluate augmentation effectiveness. Specifically, $AP_{50}$ (a strong indicator of good localization and classification scores) is calculated with an IoU threshold of 0.5 for each category, and averaged for $mAP_{50}$. Additionally, we use FID (Heusel et al., 2017) to assess the overall realism of synthetic images, and DINOv2 Score to evaluate the foreground fidelity.

**Methods for comparison.** We compare three types of data augmentation methods. (1) Copy & Paste: A simple method by copying abnormal cells from the abnormal cell bank, resizing and pasting them onto the specified locations in the background images. (2) Generation-based method: We adopt GLIGEN (Li et al., 2023), an advanced diffusion

Table 1: Performance comparison of different methods (**Best results**, second best results).

| Detector | Method | mAP$_{50}$ | Tail | | | | Non-tail | | | | | | |
|---|---|---|---|---|---|---|---|---|---|---|---|---|---|
| | | | flora | actin | herps | cand | lsil | ascus | scc | asch | agc | trich | hsil |
| YOLOv8 | Baseline | 51.7 | 68.9 | 64.3 | 83.6 | 54.7 | 43.8 | 28.0 | 21.1 | **19.8** | 66.3 | **68.4** | 50.1 |
| | Copy & Paste | 52.1 | 72.5 | 68.9 | **86.7** | 48.3 | 43.7 | 28.5 | **22.9** | 18.2 | 66.2 | 66.7 | 50.4 |
| | GLIGEN (CVPR 2023) | 51.8 | 77.6 | 65.6 | 79.9 | **54.8** | 41.2 | 30.9 | 20.3 | 17.6 | 66.2 | 66.7 | 49.2 |
| | Paint-by-Example (CVPR 2023) | 48.7 | 64.6 | 62.9 | 79.9 | 44.3 | 40.9 | 30.9 | 20.4 | 15.2 | 63.2 | 65.0 | 48.5 |
| | ObjectStitch (CVPR 2023) | 50.2 | 69.9 | 66.0 | 84.8 | 47.0 | 40.4 | 30.0 | 17.6 | 17.3 | 64.3 | 65.2 | 49.3 |
| | **SAIC (Ours)** | **54.8** | **84.4** | **77.5** | 85.8 | 51.7 | **44.3** | **31.5** | 22.8 | 19.6 | **67.6** | 66.9 | **50.7** |
| Faster R-CNN | Baseline | 59.4 | 72.8 | 78.9 | 83.5 | 68.8 | **60.7** | 43.0 | 31.8 | **23.7** | 69.6 | 67.9 | 52.6 |
| | Copy & Paste | 59.5 | 70.5 | 77.6 | 83.6 | 73.5 | 59.9 | 44.7 | 35.0 | 22.8 | 68.3 | 66.8 | 51.3 |
| | GLIGEN (CVPR 2023) | 59.5 | 76.6 | 77.4 | 82.0 | **79.5** | 57.3 | 42.5 | 32.9 | 19.8 | 69.1 | 66.1 | 50.9 |
| | Paint-by-Example (CVPR 2023) | 59.1 | 81.4 | 74.7 | 77.2 | 72.3 | 58.2 | 43.8 | 31.0 | 22.7 | 67.6 | **68.4** | 52.7 |
| | ObjectStitch (CVPR 2023) | 59.7 | 77.1 | 82.5 | 82.9 | 68.2 | 55.6 | 44.1 | 34.8 | 21.9 | **70.4** | 65.8 | **52.9** |
| | **SAIC (Ours)** | **61.9** | **83.7** | **85.6** | **85.9** | 76.3 | 59.4 | **44.8** | **36.9** | 22.9 | 68.4 | 65.2 | 52.3 |

model for image inpainting, following the setting of (Shen et al., 2024), we fine-tuned on the Comparison detector Database for 50k iterations. (3) Composition-based methods: We include Paint-by-Example (Yang et al., 2023) and ObjectStitch (Song et al., 2023), two mainstream models that support the same input format as ours and require no additional parameter fine-tuning. Note that since data-augmented methods are utilized during the model training phase (to generate enriched synthetic data for training), they do not alter the inference time or memory consumption of the anomaly cell detection model to influence the practical deployment.

## 3.2. Validation of Data Augmentation Effectiveness

We validate the effectiveness of our SAIC through its improvement in detector performance and complement to staining styles in the training set.

**Improvement in detector performance.** Under identical experimental conditions of adding 5,696 synthetic images into the initial training set, we compared the performance improvements achieved by different augmentation methods. As shown in Table 1, SAIC yields the best average performance improvement across both detection models, enhancing YOLOv8 by 3.1 points and Faster R-CNN by 2.5 points. This method is especially effective for tail categories. For example, flora achieves improvements of 15.5 points for YOLOv8 and 10.9 points for Faster R-CNN; and actinomyces (actin) gain 13.2 points for YOLOv8 and 6.7 points for Faster R-CNN.

**Complement to staining styles.** For the four tail categories, we use color histograms to roughly represent their staining styles and perform t-SNE analysis on style distributions of their training sets, augmented data, and test sets. As shown in Figure 3, SAIC effectively complements the staining styles distribution in the training set, thereby enhancing the detector's robustness to staining variations. More investigations of data augmentation effectiveness are shown in Appendix C.

## 3.3. Quality Evaluation of Augmented Image

We evaluate the quality of augmented images synthesized by SAIC through qualitative comparisons, quantitative comparisons, and a user study.

**Qualitative comparisons.** As shown in Figure 4, Copy & Paste produces synthetic images with a low informational density as it does not introduce new information. The

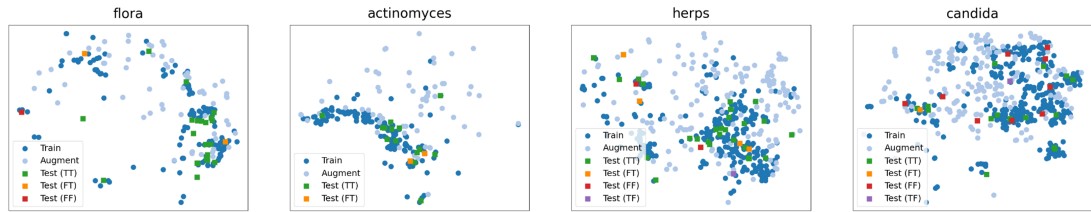

Figure 3: t-SNE analysis of staining style complement on tail categories. The first/second value of legends indicates the detection state before/after augmentation, and T/F represents the true/false detection.

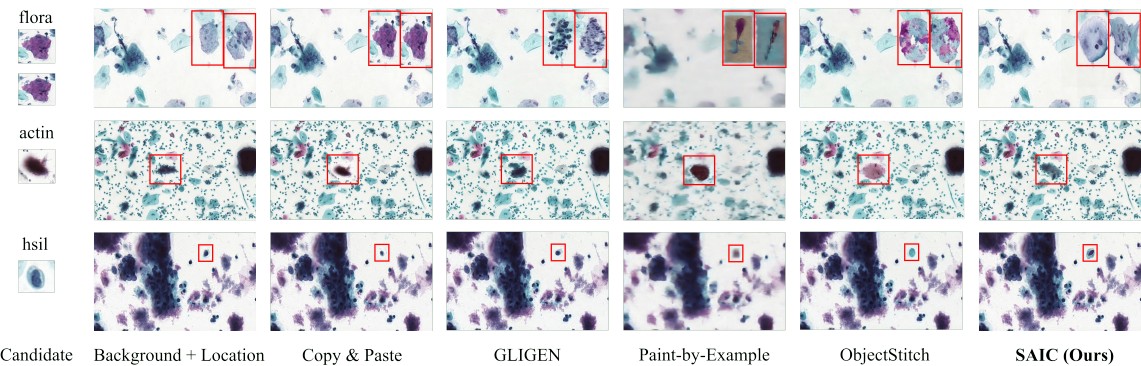

Figure 4: Qualitative comparisons across augmentation methods on flora, actin (top-2 tail categories) and hsil (top-1 non-tail category).

Generation-based method GLIGEN (Li et al., 2023) generates visually realistic images but struggles with tail category representation and fidelity to real cells due to diffusion models' bias toward simpler, in-distribution samples. In contrast, our SAIC synthesizes images with high fidelity and rich informational density, outperforming other Composition-based methods. More examples are shown in Appendix D.

**Quantitative comparisons.** As shown in Table 2, compared to other composition-based methods, our SAIC significantly excels in the overall realism of synthetic images (indicated by FID), and achieves the highest foreground fidelity (indicated by DINOv2 Score). Although the fine-tuned GLIGEN (Li et al., 2023) offers marginally better overall realism, it falls short of preserving the fidelity of candidate cells used as the foreground, which limits its effectiveness for downstream tasks. In contrast, our SAIC provides a balanced performance, excelling in both aspects.

**User study.** We conduct a user study involving 8 experienced pathologists to evaluate the quality of SAIC-synthesized images. Each pathologist was required to assess 50 images (25 synthetic and 25 real) for their realism within a 30-minute timeframe. According to the mean and standard deviation results presented in Table 3, the average accuracy of distinguishing between real and synthetic images was 50%. And the judgment distributions

of actual real and synthetic images are consistent. The results demonstrate that the fidelity of the synthetic images is sufficient to deceive human observers and reaffirm their high quality in supporting cytopathological diagnostics.

Table 2: Quantitative comparisons.

| Framework | Method | FID ↓ | DINOv2 Score ↑ |
|---|---|---|---|
| Generation | GLIGEN | **9.1** | 81.6 |
| Composition | Paint-by-Example | 191.0 | 74.3 |
| Composition | ObjectStitch | 95.6 | 79.2 |
| Composition | **SAIC (Ours)** | 9.7 | **86.5** |

Table 3: User study results.

| | Pred. Real | Pred. Syn | Total |
|---|---|---|---|
| Real | 14.875±3.295 | 10.125±3.295 | 25 |
| Syn | 14.875±2.976 | 10.125±2.976 | 25 |
| Total | 29.750±5.449 | 20.250±5.449 | 50 |

Table 4: Ablation study on impacts of various strategies.

| Attribute-based Selection | | | Style-aligned Composition | | LVLM-based | $mAP_{50}$ ↑ | | FID ↓ | DINOv2 Score ↑ |
|---|---|---|---|---|---|---|---|---|---|
| Category | Area | Type | Self | Background | Filtration | YOLOv8 | Faster R-CNN | | |
| | | | ✓ | | | 51.5 | 58.4 | 12.0 | 86.8 |
| | | | | ✓ | | 50.6 | 58.8 | 10.5 | 85.2 |
| ✓ | | | ✓ | | | 53.2 | 60.3 | 11.4 | 86.4 |
| ✓ | ✓ | | ✓ | | | 53.6 | 60.5 | 10.6 | **87.0** |
| ✓ | ✓ | ✓ | | | | 53.8 | 60.6 | 10.5 | 86.1 |
| ✓ | ✓ | ✓ | ✓ | | | 53.9 | 60.9 | 10.2 | 86.9 |
| ✓ | ✓ | ✓ | | ✓ | | 54.1 | 61.3 | **9.5** | 85.8 |
| ✓ | ✓ | ✓ | ✓ | ✓ | ✓ | **54.8** | **61.9** | 9.7 | 86.5 |

### 3.4. Ablation Study

We conduct comprehensive ablation studies to validate the effectiveness of the various steps employed in SAIC. The comparison results presented in Table 4 confirm that our designed steps improve the composition quality and significantly enhance detection performance.

Specifically, Attribute-based Selection, despite its simplicity, yields robust improvements, and incorporating self- and background-style-aligned composition both leads to further performance boosts, where background-style alignment yields more pronounced enhancements compared to self-style alignment. Besides, self-style alignment prioritizes fidelity (higher DINOv2 scores), while background-style alignment favors realism (lower FID). Critically, integrating both mechanisms through LVLM-based Filtration synergizes their strengths, achieving superior overall performance. Qualitative results of the ablation study are provided in Appendix E.

### 3.5. Cross Domain Application

We demonstrate SAIC's generalizability by presenting a comparative evaluation of SAIC against baseline methods for synthesizing cells in three external pathological image types: circulating tumor cells (CTC), blood cells (BC), and urine cells (UC). The baselines include both the basic GLIGEN and the version of fine-tuning on the Comparison Detector Database (denoted as GLIGEN-base and GLIGEN-ft, respectively). As shown in Figure 5, SAIC consistently achieves superior fidelity and style coherence across all three cytopathological image synthesis tasks, thereby highlighting its potential for broader application in cytopathological image data augmentation.

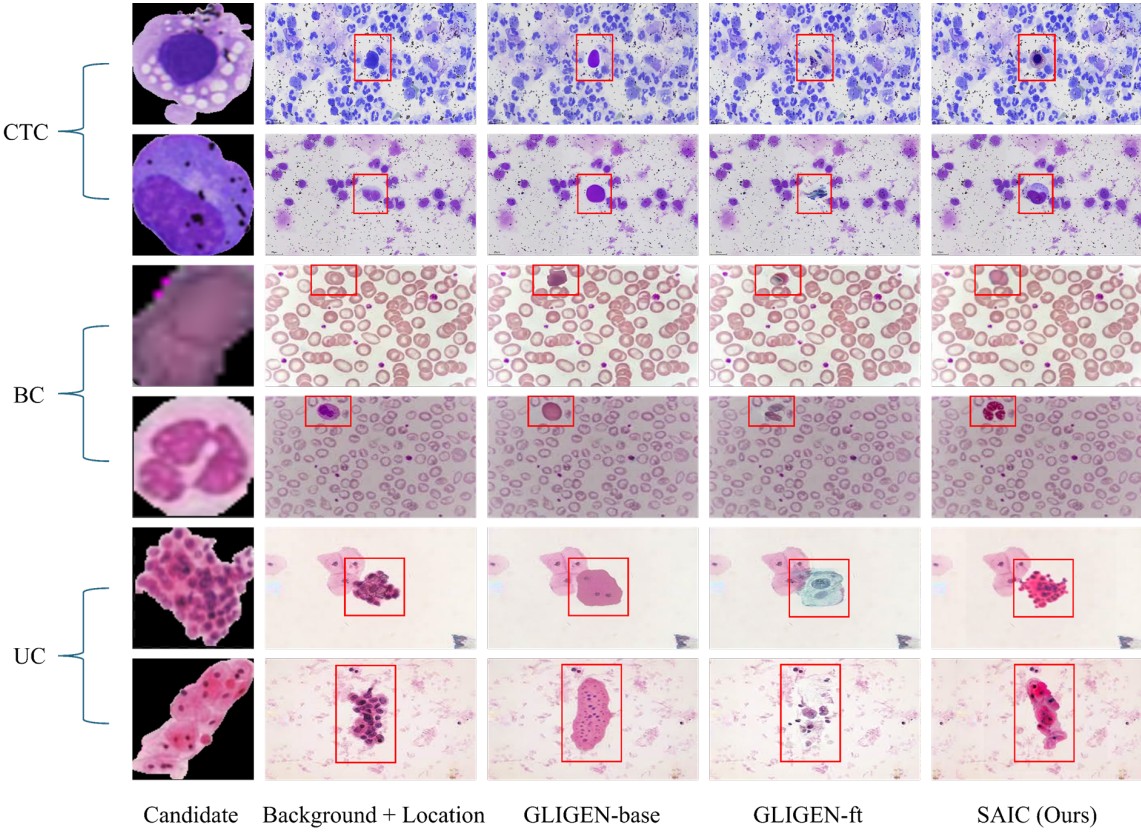

Figure 5: Qualitative comparisons in three external pathological image types.

## 4. Conclusion

This paper proposes **S**tyle **A**ligned **I**mage **C**omposition (SAIC), a training-free data augmentation architecture for cytopathological abnormal cell detection, to address issues of limited, long-tailed distributions and biased staining styles in pathological image data. By introducing *Attribute-based Selection*, *Style-aligned Composition*, and *LVLM-based Filtration*, SAIC achieves high-fidelity and style-preserved data augmentation. Experimental results demonstrate that, compared to the existing data augmentation methods, SAIC-synthesized data more effectively enhances the performance and robustness of abnormal cell detection models for pathological images, showing notable advantages for tail categories. Moreover, SAIC exhibits outstanding fidelity and generalizability. This framework provides a universal data augmentation solution for cytopathological images and can potentially impact various cross-domain applications positively.

## Acknowledgments

This work is supported by the Natural Science Foundation of China under Grant 62271465, Suzhou Basic Research Program under Grant SYG202338, and Open Fund Project of Guangdong Academy of Medical Sciences, China (No. YKY-KF202206).

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

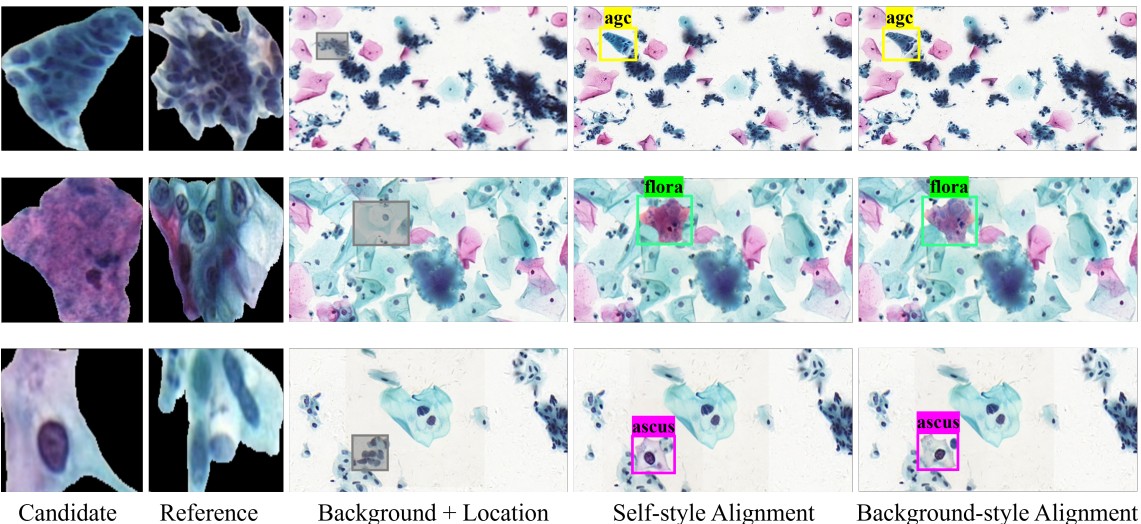

Candidate · Reference · Background + Location · Self-style Alignment · Background-style Alignment

Figure 6: Demonstration of staining style alignment.

## Appendix A.  Staining Style Alignment

Figure 6 shows the composition results with self- and background-style alignment. Through self-style alignment, synthetic cells can effectively retain their own staining styles. However, this may result in inconsistencies with the staining style of the background image. By applying background-style alignment, synthetic cells can integrate more harmoniously into the background.

Note that we do not simply apply background-style alignment. The main reason is that, given the limited size of the abnormal cell bank, the reference cells selected using the DINOv2 Score may still be inconsistent with the staining style of the background image. In such cases, applying background-style alignment could negatively impact the fidelity of synthetic images.

## Appendix B.  LVLM-based Filtration

Figure 7 shows the detailed prompt setting of LVLM-based Filtration. Leveraging the perception of large vision-language models like GPT-4 for images and their understanding of text, we can design appropriate prompts to enable them to automatically filter the more harmonized one from two synthetic images and provide comprehensive reasons.

Note that in the subsequent experiments, through LVLM-based filtration, 64% of the synthetic images produced by our SAIC are derived from background-style alignment, while 36% are derived from self-style alignment.

## Appendix C.  More Investigations of Data Augmentation Effectiveness

**Effectiveness of data augmentation across scaling ratios.** To validate the effectiveness of SAIC in relieving the practical puzzle of large-scale training dataset collection,

---

**Fidelity filtration based on GPT-4**

**Question:** Could you select the one with higher fidelity from the following two images? You could analyze them in terms of how well the cells in bounding box harmonize with the background.

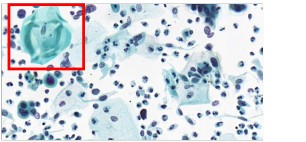 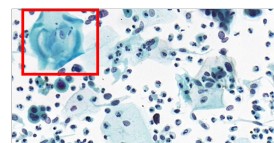

**Answer:** Yes, I would select the second image. The reasons are as follows:
1. In the first image, the cells within the bounding box appear to be a more saturated blue-green color, which is more intense and contrasts markedly with the other cells in the background.
2. In the second image, the cells within the bounding box are slightly softer in color, and although still blue-green, they are less saturated and closer to the color of the background.

In conclusion, There is a better blend between the cells and the background in the second image, resulting in higher overall fidelity.

Figure 7: Demonstration of LVLM-based Filtration.

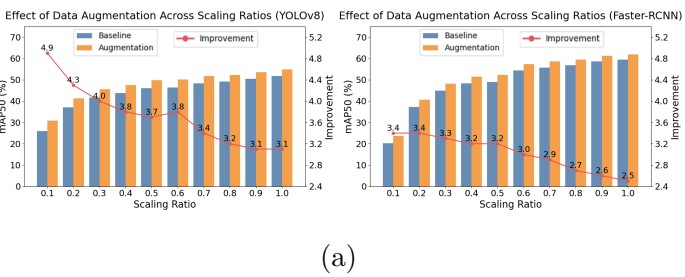 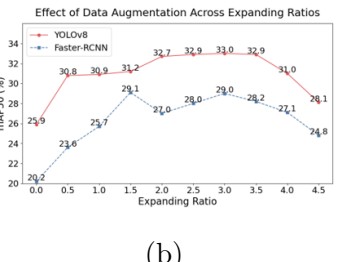

(a)            (b)

Figure 8: More investigations of data augmentation effectiveness.

we evaluate our framework's performance with varying amounts of initial training data. Reduced training sets are sampled from the original data at proportions of 0.1 to 0.9, maintaining class distribution, and consistent data augmentation is applied to each reduced set. As shown in Figure 8 (a), our SAIC significantly enhances detector performance compared to the baseline. However, as the initial training data increases, the baseline accuracy and the improvement from augmentation gradually converge, likely due to the limited diversity of the data distribution.

**Effect of data augmentation across expanding ratios.** We also evaluate the impact of different degrees of data augmentation on the training set sampled from the original data at the proportion of 0.1. As shown in Figure 8 (b), the improvement on detector performance initially improves but declines as the degree of augmentation increases. This decline occurs because excessive synthetic data skews the training distribution, reducing alignment with real test data.

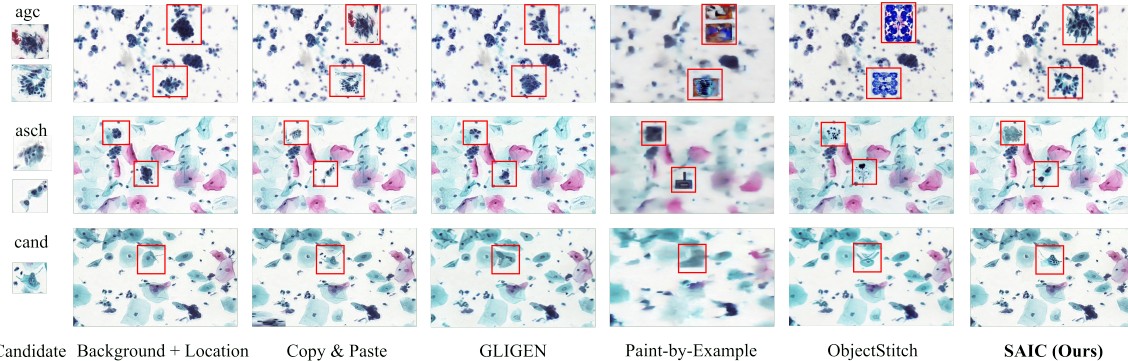

Figure 9: Qualitative comparisons across augmentation methods on agc, asch, and cand.

## Appendix D. More Qualitative Comparisons

Figure 9 shows more examples of qualitative comparisons. For candidate cells with various attributes (category, type, and area) and different background images, our SAIC consistently synthesizes images with high fidelity and rich informational density, outperforming other augmentation methods.

## Appendix E. Ablation Study

Figure 10 shows the qualitative results of the ablation study. The first and last columns show the real background image (Real) and the synthesis results of the full framework (Full), respectively. The intermediate columns illustrate the effects of omitting core strategies: (1) None: No strategies applied, resulting in random synthesis with disrupted cell distribution patterns; (2) w/ Stage 1: Attribute-based Selection strategy applied alone, yielding decent synthesis quality; (3) w/ Stage 2: Style-aligned Composition strategy applied alone, resulting in a lack of harmony between the foreground and background.

Note that in our experiments, through LVLM-based filtration, 64% of the synthetic images produced by our SAIC are derived from background-style alignment, while 36% are derived from self-style alignment.

## Appendix F. Computational Efficiency Analysis

We emphasize that since augmented data is utilized during the model training phase (to generate enriched synthetic data for training), it does not alter the inference time or memory consumption of the anomaly cell detection model. Given that this task does not impose stringent real-time requirements, the computational efficiency of YOLOv8 and Faster R-CNN is sufficient for practical deployment. Furthermore, we compared the average time and memory consumption for augmented image generation, as summarized in Table 5. SAIC achieves a generation speed of 12.81 seconds per image, with average time allocations of 0.04s, 8.79s, and 3.98s for the selection, composition, and filtration stages, respectively. While SAIC's generation time is marginally longer than baseline methods (e.g., GLIGEN), it

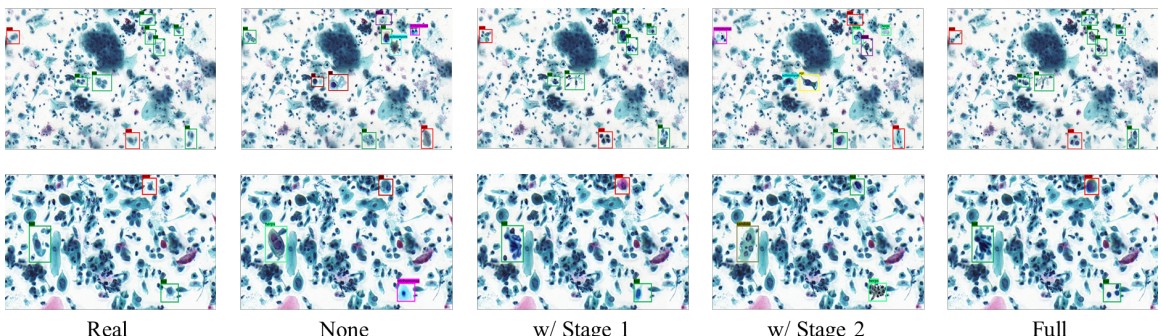

| Real | None | w/ Stage 1 | w/ Stage 2 | Full |

Figure 10: Qualitative results of the ablation study.

Table 5: Comparison of time cost and memory usage across augmentation methods.

| Framework | Method | Average Time (second per image) | Memory Usage (MiB) |
|---|---|:---:|:---:|
| Generation | GLIGEN | 10.20 | 16225 |
| Composition | Paint-by-Example | 4.60 | 12161 |
| Composition | ObjectStitch | 4.30 | 11883 |
| Composition | SAIC (Ours) | $0.04 + 8.79 + 3.98 = 12.81$ | 12657 |

eliminates the need for supervised fine-tuning on domain-specific data and delivers superior synthesis quality. Lastly, SAIC's memory usage remains comparable to baseline approaches.

## Appendix G. Comparison with Model-based Methods

We have introduced two model-based methods for long-tailed object detection performance comparison:

1. Faster R-CNN (RS): A Faster R-CNN model trained with resampling, a common solution for long-tailed problems.

2. BACL: A data-balancing method from *Balanced Classification: A Unified Framework for Long-Tailed Object Detection* (TMM 2023).

It is worth noting that, as an augmentation-based method, SAIC and the model-based methods are mutually compatible. We further demonstrate the synergistic effects of combining SAIC with these approaches. As shown in Table 6, SAIC outperforms both baseline methods in improving anomaly detection performance for both overall and tail categories. This indicates that enhancing tail-class diversity via SAIC provides more substantial gains compared to the re-weighting strategies employed by model-based methods. Moreover, integrating SAIC with model-based methods yields additional performance improvements in abnormal cell detection, underscoring the complementary nature of these two methodological paradigms.

Table 6: Comparison with model-based methods (**Best results**, second best results).

| Method | mAP$_{50}$ | Tail | | | |
| --- | --- | --- | --- | --- | --- |
| | | **Flora** | **Actin** | **Herps** | **Cand** |
| Faster R-CNN | 59.4 | 72.8 | 78.9 | 83.5 | 68.8 |
| Faster R-CNN (RS) | 59.8 | 74.7 | 80.6 | 84.3 | 70.9 |
| BACL | 60.7 | 76.5 | 81.3 | 85.2 | 72.1 |
| Faster R-CNN + SAIC | 61.9 | 83.7 | 85.6 | 85.9 | 76.3 |
| Faster R-CNN (RS) + SAIC | 62.2 | 84.1 | 86.2 | 86.3 | 77.3 |
| BACL + SAIC | **62.6** | **84.3** | **86.5** | **86.4** | **77.5** |

## Appendix H. Statistical Tests

We conduct bootstrap resampling tests comparing SAIC with baseline methods and the best-performing comparative methods across two detection models (Copy & Paste for YOLOv8 and ObjectStich for Faster R-CNN). The results, summarized in Table 7, demonstrate statistically significant differences between SAIC and all compared methods (p-value <0.05).

Table 7: Statistical tests.

| Detector | Method | p-value |
| --- | --- | --- |
| YOLOv8 | Baseline | 0.028 |
| | Copy & Paste | 0.039 |
| Faster R-CNN | Baseline | 0.031 |
| | ObjectStitch | 0.036 |

