# OpenReview forum: "Style-Aligned Image Composition for Robust Detection of Abnormal Cells in Cytopathology"
_MIDL.io/2025/Conference — MIDL 2025 Oral_

### Official Review · Reviewer_e1nm · 2025-02-21

**Confidence:** 3
**Preliminary Rating:** 4
**Recommendation:** Poster
**Final Rating:** 4

**Summary:**

The paper tackles the challenge of improving abnormal cell detection in cytopathology, where limited annotations, long-tailed cell distributions, and inconsistent staining styles hinder robust model training. It asks how to effectively augment data so that synthetic images preserve both the identity of abnormal cells and the staining style of their background, especially for underrepresented (tail) categories. To address these questions, the authors introduce a training-free SAIC framework that selects candidate abnormal cells based on their attributes, employs high-frequency feature reconstruction for style alignment, and uses a large vision-language model to filter the best-quality synthetic images. This design directly targets the issues of data scarcity and style inconsistency, ultimately enhancing the detection performance and robustness of deep learning models in clinical applications.

**Strengths:**

1. It is a training-free pipeline that leverages large foundation models to improve the robustness of cell detection and classification.
2. The approach achieves high-fidelity and style-preserved synthesis by aligning the abnormal cell’s features with the background’s staining style.
3. Incorporating SAIC-synthesized images improves abnormal cell detection performance, especially for tail categories.

**Weaknesses:**

1. The authors’ claims of broad generalizability and robustness across cross-domain applications are only briefly illustrated (e.g., in Appendix F) and might benefit from further, more extensive validation in diverse real-world scenarios.
2. The framework integrates multiple pre-trained models (SAM, DINOv2, ControlNet, Stable Diffusion) and employs a multi-stage pipeline (attribute-based selection, style-aligned composition, and LVLM-based filtration). The time and step complexity may hinder reproducibility on external datasets.

**Detailed Comments:**

What is the memory usage and model complexity of the design compared with other trainable methods? What is the time cost comparison?

**Justification Of The Final Rating:**

I am satisfied with the additional experimental results and clarifications provided by the author in the rebuttal, which have improved the quality of the paper and addressed all my concerns. Therefore, I recommend accepting the paper.

**Justification Of The Preliminary Rating:**

The preliminary rating is justified by the paper’s strong contributions to a challenging problem. The proposed training-free SAIC framework is innovative in leveraging multiple pre-trained models to generate high-fidelity, style-preserved synthetic images that improve abnormal cell detection performance, especially for underrepresented tail categories. Quantitative results, ablation studies, and user evaluations provide solid evidence supporting these contributions.

**Questions To Address In The Rebuttal:**

Adding a discussion of cross-domain applications and providing efficiency results (e.g., memory usage, model complexity, and time cost) may offer a better understanding of the model.

---

> ### Author Response · Authors · 2025-03-07
> **Response to the Comments of Reviewer e1nm**
>
> **Comment 1: Add a discussion of cross-domain applications.**
>
> Thank you for the constructive feedback. To better demonstrate SAIC’s cross-domain applications, **we have expanded the analysis in Appendix F and incorporated them in Section 3.5**. As illustrated in Figure 5, we present a comparative evaluation of SAIC against baseline methods for synthesizing cells in three external pathological image types: circulating tumor cells (CTC), blood cells (BC), and urine cells (UC). The baselines include both the basic GLIGEN and the version of fine-tuning on the Comparison Detector Database (denoted as GLIGEN-base and GLIGEN-ft, respectively). The results demonstrate that SAIC consistently achieves superior fidelity and style coherence across all three cytopathological image synthesis tasks, thereby highlighting its potential for broader application in cytopathological image data augmentation.
>
> **Comment 2: Provide efficiency results.**
>
> Thank you for your comment. We emphasize that since augmented data is utilized during the model training phase (to generate enriched synthetic data for training), **it does not alter the inference time or memory consumption of the anomaly cell detection model.** Given that this task does not impose stringent real-time requirements, the computational efficiency of YOLOv8 and Faster R-CNN is sufficient for practical deployment. Thanks again for the constructive suggestion. **We supplement the clarification about this issue in the *Methods for comparison* part of Section 3.1.**
>
> Furthermore, we compared the average time and memory consumption for augmented image generation, as summarized in the table below. SAIC achieves a generation speed of 12.81 seconds per image, with average time allocations of 0.04s, 8.79s, and 3.98s for the selection, composition, and filtration stages, respectively. While SAIC’s generation time is marginally longer than baseline methods (e.g., GLIGEN), it eliminates the need for supervised fine-tuning on domain-specific data and delivers superior synthesis quality. Lastly, SAIC’s memory usage remains comparable to baseline approaches. We supplement the discussion of time and memory consumption for augmented image generation in Appendix F.
> | Framework | Method            | Average Time (second per image) | Memory Usage (MiB) |
> |----------|--------------------|:--------------------------------:|:------------------:|
> |Generation| GLIGEN            | 10.20                           | 16225             |
> |Composition| Paint-by-Example  | 4.60                            | 12161             |
> |Composition| ObjectStitch      | 4.30                            | 11883             |
> |Composition| SAIC (Ours)             | 0.04 + 8.79 + 3.98 = 12.81   | 12657             |

---

### Official Review · Reviewer_4YWX · 2025-02-21

**Confidence:** 4
**Preliminary Rating:** 5
**Recommendation:** Best Paper Award, Oral
**Final Rating:** 5

**Summary:**

The paper introduces a training-free method for synthesizing realistic pathological images by aligning abnormal cell styles with the background. This improves segmentation robustness, especially for rare cells. The approach enhances detection performance and adapts to different pathological domains without extra training in style-aligned phase.

**Strengths:**

The paper is very well-written and well-structured, presenting clear experiments that support the proposed method’s effectiveness.

The paper presents a training-free approach for generating realistic pathological images, making it highly adaptable and efficient. By aligning abnormal cell styles with the background, the method enhances segmentation robustness. It effectively addresses staining inconsistencies and data imbalance, which are notorious challenges in cytopathology. The proposed approach outperforms traditional augmentation techniques without requiring additional training, making it computationally efficient. Moreover, the method’s ability to generalize across different pathological domains increases its practical applicability. The integration of a vision-language model for filtering synthesized images adds another layer of refinement.

**Weaknesses:**

The paper lacks extensive validation across multiple datasets (as mentioned in the manuscript).

The paper remains unclear how much improvement is directly attributable to style alignment versus the combined impact of all components.

No statistical tests confirm significance.

Reliance on a vision-language model may introduce bias.

**Detailed Comments:**

The paper is very well written and presents a method that could be extended in various directions. While an ablation study is included, it remains unclear how much improvement is directly due to style alignment versus the combined effect of all components.

When working with large datasets, VLM outputs may vary depending on model design, prompt phrasing, and dataset biases. To streamline and automate the pipeline, a structured approach could be discussed.

**Justification Of The Final Rating:**

My comments have been well addressed, and the limitations are clearly explained. The revisions offer a thorough clarification, enhancing the precision and depth of the discussion while ensuring a more comprehensive understanding.

**Justification Of The Preliminary Rating:**

The paper is well-written and presents a promising approach, but the direct impact of style alignment remains unclear. Statistical validation is lacking, and reliance on a vision-language model may introduce bias.

**Questions To Address In The Rebuttal:**

The ablation study is performed, but how much improvement is directly attributable to style alignment rather than the combined effect of all components?

Could additional experiments be conducted to further isolate the contribution of style alignment?

Are the reported improvements statistically significant?

Were any external datasets used to validate performance beyond the reported experiments?

Could the use of VLMs introduce bias or inconsistencies, and how can this be mitigated?

**Special Issue:**

Yes

---

> ### Author Response · Authors · 2025-03-07
> **Response to the Comments of Reviewer 4YWX**
>
> **Q1: The ablation study is performed, but how much improvement is directly attributable to style alignment rather than the combined effect of all components? Could additional experiments be conducted to further isolate the contribution of style alignment?**
>
> We appreciate the valuable suggestion. We realized that the original ablation study did not include the results that fully isolate the contribution of style alignment. **In response, we have revised the presentation of Table 4 and supplemented the additional results (see Row 5)**. By comparing the outcomes of self-style alignment and background-style alignment (Rows 6 and 7), we demonstrate that both variants of style alignment contribute to performance improvements. Furthermore, the introduction of LVLM-based filtration to achieve adaptive style alignment (Row 8) leads to additional gains. These results explicitly quantify the distinct contributions of style alignment mechanisms.
>
> **Q2: Are the reported improvements statistically significant?**
>
> Thank you for the constructive comment. To rigorously evaluate statistical significance, we conducted bootstrap resampling tests comparing SAIC with baseline methods and the best-performing comparative methods across two detection models (*Copy & Paste* for YOLOv8 and *ObjectStich* for Faster R-CNN). The results, summarized in the table below, demonstrate statistically significant differences between SAIC and all compared methods (p-value < 0.05). **We have supplemented this analysis in Appendix H to provide full transparency**. We sincerely appreciate your suggestion, which has strengthened the statistical validity of our findings.
> | Detector      | Method         | p-value |
> |--------------|---------------|:-------:|
> | YOLOv8   | Baseline       |  0.028  |
> |              | Copy & Paste   |  0.039  |
> | Faster R-CNN | Baseline  |  0.031  |
> |              | ObjectStitch   |  0.036  |
>
> **Q3: Were any external datasets used to validate performance beyond the reported experiments?**
>
> Thank you for your valuable suggestion. Unfortunately, we have not yet validated performance using external datasets due to the limited availability of large-scale, open-source cytopathological image datasets, which precludes supplementary external validation in the short term. However, **we provide qualitative comparative results in Section 3.5 and Figure 5** of SAIC and baselines for synthesizing pathological images of *external cell types* (circulating tumor cells, blood cells, and urine cells), demonstrating SAIC’s robust generalizability in pathological image synthesis. We plan to further investigate SAIC’s performance in abnormal cell detection across additional pathological image categories in future work.
>
> **Q4: Could the use of VLMs introduce bias or inconsistencies, and how can this be mitigated?**
>
> Thank you for your thoughtful comments. A key potential bias in VLMs is positional bias, where the model may disproportionately favor the first (or second) option in a binary choice setup. To mitigate this, we systematically shuffled the order of the two choices during experiments to eliminate positional preference. Thanks again for your suggestion. **We supplemented the clarification about this issue in Section 2.3.**
>
> Furthermore, as shown in Rows 6 and 7 of revised Table 4, even if the filtration step degenerates into exclusively selecting either self-style-aligned or background-style-aligned images, SAIC still retains significant performance improvements. This indicates that **the lower bound of bias introduced by the LVLM is well-controlled**.
> Additionally, advanced strategies such as Mixture of Experts (MoE) frameworks could further reduce potential biases or inconsistencies from LVLM-based filtration. We appreciate this suggestion and plan to explore such enhancements in future studies.

---

### Official Review · Reviewer_x2vC · 2025-02-27

**Confidence:** 5
**Preliminary Rating:** 4
**Final Rating:** 4

**Summary:**

This paper introduces Style Aligned Image Composition (SAIC), a training-free data augmentation framework designed to improve abnormal cell detection in cytopathological images. SAIC addresses challenges such as limited datasets, long-tailed distributions, and staining style biases by integrating Attribute-based Selection, Style-aligned Composition, and LVLM-based Filtration to generate high-fidelity, style-consistent synthetic data. Experimental evaluations show that SAIC outperforms existing augmentation techniques, significantly boosting model performance and robustness, especially for underrepresented categories.

**Strengths:**

The paper demonstrates several key strengths, making it a valuable contribution to the field of cytopathological image analysis. First, it introduces Style Aligned Image Composition (SAIC) as a training-free data augmentation method. Unlike conventional augmentation techniques, SAIC directly addresses critical challenges such as staining style bias, long-tailed distributions, and limited data availability, which are common in medical imaging.

Second, the paper presents well-executed experiments that demonstrate the impact of SAIC on abnormal cell detection models. By showing improved generalizability and fidelity, the study suggests that SAIC can be applied to broader cross-domain tasks beyond cytopathology.

Third, the paper is well-structured, clearly written.

**Weaknesses:**

This study tackles challenges such as long-tailed data distributions and inconsistent staining styles by introducing a novel data augmentation approach. Unlike methods that address these issues during model training, this approach focuses on data pre-processing. However, a direct comparison between the proposed method and model-based approaches would strengthen the study, particularly in terms of quantitative performance and computational efficiency. Including such a comparison would provide deeper insights into the trade-offs between augmentation and training-based solutions.

Regarding the Style Map Extraction step, the generation of two types of high-frequency maps appears redundant, potentially introducing unnecessary complexity. Additionally, in the LVLM-based Filtration step, the prompts seem biased toward harmonized results, which could explain why 64% of SAIC-generated images align with background styles. A more unified approach that integrates self- and background-style alignment could eliminate the need for LVLM-based Filtration, simplifying the pipeline while maintaining effectiveness.

**Detailed Comments:**

Here are some additional comments and suggestions for minor improvements and clarifications in the paper:

1.	Clarification on SAIC’s Generalizability – While the paper claims that SAIC has strong generalizability, it would be helpful to provide more discussion on its applicability beyond cytopathological images.

2.	Computational Efficiency Analysis – The study would benefit from a clearer breakdown of the computational cost of SAIC compared to existing augmentation techniques. Including inference time, memory consumption, or complexity analysis would help assess its practicality in real-world settings.

3.	Justification for High-Frequency Maps – The generation of two high-frequency maps in the Style Map Extraction step seems redundant. Providing more explanation or an ablation study comparing a single high-frequency map approach to the proposed method would clarify its necessity.

4.	Quantitative Comparisons with Model-based Methods – While SAIC is a pre-processing method, a direct comparison with model-based techniques that address the same challenges would strengthen the paper. The authors could provide quantitative results for both approaches in terms of accuracy, robustness, and computational requirements.

**Justification Of The Final Rating:**

The paper is well-written and presents a promising approach. Although its generation time is slightly longer than that of other methods, the use of a training-free method for synthesizing realistic pathological images remains highly appealing.

**Justification Of The Preliminary Rating:**

The proposed Style Aligned Image Composition (SAIC) is an innovative, training-free data augmentation framework that effectively addresses long-tailed distributions and staining style biases in cytopathological images.

The paper is well-structured. The experimental results demonstrate clear improvements over existing augmentation techniques, particularly in tail categories. Additionally, SAIC’s potential generalizability to other medical imaging domains adds further value.

**Questions To Address In The Rebuttal:**

Addressing these points—especially providing quantitative comparison with model-based approaches, clearer justification for the necessity of generating two high-frequency maps, and further discussion on computational efficiency —would significantly strengthen the paper’s claims.

---

> ### Author Response · Authors · 2025-03-07
> **Responses to Comments 1-3 of Reviewer x2vC**
>
> **Comment 1: Clarification on SAIC’s Generalizability.**
>
> Thank you for the constructive feedback. To better demonstrate SAIC’s generalizability, **we have expanded the analysis in Appendix F and incorporated them in Section 3.5**. As illustrated in Figure 5, we present a comparative evaluation of SAIC against baseline methods for synthesizing cells in **three external pathological image types**: circulating tumor cells (CTC), blood cells (BC), and urine cells (UC). The baselines include both the basic GLIGEN and the version of fine-tuning on the Comparison Detector Database (denoted as GLIGEN-base and GLIGEN-ft, respectively). The results demonstrate that SAIC consistently achieves superior fidelity and style coherence across all three cytopathological image synthesis tasks, thereby highlighting its potential for broader application in cytopathological image data augmentation.
>
> **Comment 2: Computational Efficiency Analysis**
>
> Thank you for your comment. We emphasize that since augmented data is utilized during the model training phase (to generate enriched synthetic data for training), **it does not alter the inference time or memory consumption of the anomaly cell detection model.** Given that this task does not impose stringent real-time requirements, the computational efficiency of YOLOv8 and Faster R-CNN is sufficient for practical deployment. Thanks again for the constructive suggestion. **We supplement the clarification about this issue in the *Methods for comparison* part of Section 3.1.**
>
> Furthermore, we compared the average time and memory consumption for augmented image generation, as summarized in the table below. SAIC achieves a generation speed of 12.81 seconds per image, with average time allocations of 0.04s, 8.79s, and 3.98s for the selection, composition, and filtration stages, respectively. While SAIC’s generation time is marginally longer than baseline methods (e.g., GLIGEN), it eliminates the need for supervised fine-tuning on domain-specific data and delivers superior synthesis quality. Lastly, SAIC’s memory usage remains comparable to baseline approaches. **We supplement the discussion of time and memory consumption for augmented image generation in Appendix F.**
> | Framework | Method            | Average Time (second per image) | Memory Usage (MiB) |
> |----------|--------------------|:--------------------------------:|:------------------:|
> |Generation| GLIGEN            | 10.20                           | 16225             |
> |Composition| Paint-by-Example  | 4.60                            | 12161             |
> |Composition| ObjectStitch      | 4.30                            | 11883             |
> |Composition| SAIC (Ours)             | 0.04 + 8.79 + 3.98 = 12.81   | 12657             |
>
> **Comment 3: Justification for High-Frequency Maps.**
>
> Thank you for your feedback. As outlined in Section 2.2, the *self-style alignment* mechanism effectively preserves the morphological characteristics of candidate abnormal cells while retaining their original staining patterns. In contrast, the *background-style alignment* achieves staining style transfer for background regions while maintaining morphological fidelity. Representative outcomes of using self-style alignment only and background-style alignment only can be found in Row 6 and Row 7 of revised Table 4, respectively.
>
> The results demonstrate that even with a single high-frequency map, significant performance improvements are attainable. Notably, *background-style alignment* yields more pronounced enhancements compared to self-style alignment. Besides, *self-style alignment* prioritizes fidelity (higher DINOv2 scores), while *background-style alignment* favors realism (lower FID scores). Critically, integrating both mechanisms through *LVLM-based filtration* synergizes their strengths, achieving superior overall performance. **We have incorporated this comparative analysis into the Ablation Study (Section 3.4) for enhanced methodological clarity.**

---

> ### Author Response · Authors · 2025-03-07
> **Response to Comment 4 of Reviewer x2vC**
>
> **Comment 4: Quantitative Comparisons with Model-based Methods.**
>
> Thank you for your valuable suggestions. We have introduced two model-based methods for long-tailed object detection performance comparison:
> 1. **Faster R-CNN (RS)**: A Faster R-CNN model trained with resampling, a common solution for long-tailed problems.
> 2. **BACL**: A data-balancing method from *Balanced Classification: A Unified Framework for Long-Tailed Object Detection* (TMM 2023).
>
> It is worth noting that, as an augmentation-based method, **SAIC and the model-based methods are mutually compatible**. We further demonstrate the synergistic effects of combining SAIC with these approaches.
> As shown in the table below, SAIC outperforms both baseline methods in improving anomaly detection performance for both overall and tail categories. This indicates that **enhancing tail-class diversity via SAIC** provides more substantial gains compared to the re-weighting strategies employed by model-based methods. Moreover, integrating SAIC with model-based methods yields additional performance improvements in abnormal cell detection, underscoring the **complementary nature** of these two methodological paradigms. **We have incorporated this analysis into Appendix G to strengthen the discussion.**
> | Method                     | mAP50 | Flora | Actin | Herps | Cand  |
> |-----------------------------|:-----:|:-----:|:-----:|:-----:|:-----:|
> | Faster R-CNN                |  59.4 |  72.8 |  78.9 |  83.5 |  68.8 |
> | Faster R-CNN (RS)           |  59.8 |  74.7 |  80.6 |  84.3 |  70.9 |
> | BACL                       |  60.7 |  76.5 |  81.3 |  85.2 |  72.1 |
> | Faster R-CNN + SAIC          |  61.9 |  83.7 |  85.6 |  85.9 |  76.3 |
> | Faster R-CNN (RS) + SAIC      |  62.2 |  84.1 |  86.2 |  86.3 |  77.3 |
> | BACL + SAIC                  |  62.6 |  84.3 |  86.5 |  86.4 |  77.5 |

---

### Author Rebuttal · Authors · 2025-03-08

**Rebuttal:**

We would like to express our gratitude for the reviewers' constructive feedback. The insightful comments have substantially improved the quality of our work, particularly in clarifying methodological details and strengthening generalizability discussions.

In response to the reviewers' suggestions, we have implemented the following key revisions into the revised version:
1. Expanded Section 2.3 with enhanced clarification about VLM bias and inconsistency.
2. Revised Sections 3.1 with supplementing the clarification about computational efficiency.
3. Expanded Section 3.4 and redesigned Table 4 with a more detailed ablation study description, especially about the contribution of style alignment.
4. Expanded Section 3.5 and supplemented Figure 5 to strengthen the discussion about cross-domain validation.
5. Incorporated the supplemented results and discussions of the other comments' responses into Appendix F-H, including augmented data generation efficiency, comparison with model-based methods, and statistical test of significance.

We have systematically addressed each comment through both textual revisions and supplementary materials, with all changes highlighted in the revised manuscript using red text. Our point-by-point responses to the reviewers' concerns are provided in the separate rebuttal comments.

We remain fully available for any further clarification required and deeply appreciate the reviewers' expertise in helping strengthen this work. Thank you for your continued consideration of our manuscript.

**Supporting Material:**

/attachment/8b48b827fd006a61dc2de72f8e2f1500d2d150a6.pdf

---

### Meta-Review · Area_Chair_mpVp · 2025-03-21

**Recommendation:** Accept (Oral)
**Confidence:** 4

**Metareview:**

This paper presents a generative model-based image synthesis strategy for microscopy images. The proposed method, SAIC, is developed using components consisting of advanced but well-established frameworks such as SAM, ControlNet, Stable Diffusion, and language-vision models. The authors present a new perspective that we do not need any additional training but can instead refine the outcomes of existing models to achieve promising results for medical applications.
All the reviewers agree that the strengths outweigh the weaknesses, and so do I. I recommend acceptance of this paper.